# Analysis of Domestication Loci in Wild Rice Populations

**DOI:** 10.3390/plants12030489

**Published:** 2023-01-20

**Authors:** Sharmin Hasan, Agnelo Furtado, Robert Henry

**Affiliations:** 1Queensland Alliance for Agriculture and Food Innovation, University of Queensland, Brisbane 4072, Australia; 2Department of Botany, Jagannath University, Dhaka 1100, Bangladesh; 3ARC Centre of Excellence for Plant Success in Nature and Agriculture, Brisbane 4072, Australia

**Keywords:** wild rice, domesticated rice, domestication, polymorphisms, natural variation, genetic resources

## Abstract

The domestication syndrome is defined as a collection of domestication-related traits that have undergone permanent genetic changes during the domestication of cereals. Australian wild rice populations have not been exposed to gene flow from domesticated rice populations. A high level of natural variation of the sequences at domestication loci (e.g., seed shattering, awn development, and grain size) was found in Australian AA genome wild rice from the primary gene pool of rice. This natural variation is much higher than that found in Asian cultivated rice and wild Asian rice populations. The Australian *Oryza meridionalis* populations exhibit a high level of homozygous polymorphisms relative to domesticated rice, inferring the fixation of distinct wild and domesticated alleles. Alleles of the seed shattering genes (*SH4/SHA1* and *OsSh1*/*SH1*) present in the shattering-prone *O. meridionalis* populations are likely to be functional, while the dysfunctional alleles of these seed shattering genes are found in domesticated rice. This confirms that unlike Asian wild rice populations, Australian wild rice populations have remained genetically isolated from domesticated rice, retaining pre-domestication alleles in their wild populations that uniquely allow the impact of domestication on the rice genome to be characterized. This study also provides key information about the domestication loci in Australian wild rice populations that will be valuable in the utilization of these genetic resources in crop improvement and de novo domestication.

## 1. Introduction

Rice is one of the most important domesticated crops and a model species for cereal crops. Two cultivated rice species, namely, *Oryza sativa* L. and *Oryza glaberrima* Steud. evolved independently from the AA genome of wild *Oryza rufipogon* Griff. through the domestication process in the last 10,000 years [1,2,3]. The debate on the origin of domesticated rice remains convoluted as to whether two *Oryza* subspecies, namely, *Oryza sativa* ssp. *Japonica* and *Oryza sativa* ssp. *indica*, evolved through single or multiple domestication events. Numerous research studies based on molecular and archaeological studies have not unanimously resolved this debate. Recently, the analysis of domestication genes has gained much attention for mining more reliable information regarding domestication events [4]. In the domestication process, nucleotide polymorphisms stem from spontaneous mutations, recombination, and fixation of beneficial alleles resulting in the genetic-based morphological and physiological modifications and ecological adaptation in wild rice populations [3,5,6,7]. These modifications result in a transition from prostrate to erect growth, loss of seed shattering habit, changes in the length of awns or awnlessness, changes in hull and grain color, a decrease in seed dormancy, a low number of tillers, an increased number of grains, and diverse ecological adaptability across the geographical distributions. Apart from evolutionary mechanisms (e.g., founder events, natural selection, introgression, and genetic bottleneck), selection pressure for desirable traits by ancient humans also plays a crucial role in the domestication process [8,9]. Domesticated plants exhibit desirable domestication phenotypes, relative to their wild ancestral types. Wild progenitors demonstrate a set of undesirable characteristics, such as a prostrate plant habit, lower grain yield, inferior grain quality, non-synchronizing flowering, seed maturation, shattering, poor quality of rachis, hulls and awns, and poor culinary properties [10].

The genetic basis of domesticated loci is primarily well studied in Asian rice, as the genome of Asian rice is completely sequenced and mapped [11]. Some of the major domestication loci include *PROSTRATE GROWTH 1* (*PROG1)* [7], *LONG AND BARBED AWN 1* (*LABA1)* [12], *BROWN PERICARP AND SEED COAT* (*Rc*) [13], *Shattering (QTL)-1* (*qSH1)* [14], and *SHATTERING 4* (*SH4*) [15], whose recessive alleles contributed to the erect growth, barbless awn, white pericarp, and non-shattering seeds, respectively. Functional polymorphisms in coding sequences and promoter regions, including a 110 kb deletion in the *PROG1* gene, a 14 bp deletion in *Rc* coding sequences, and a single nucleotide change (G273T) in the *SH4* gene resulted in domestication phenotypes. Many genes including *GS3 (GRAIN SIZE 3)*, *GS5 (GRAIN SIZE 5)*, *GW2 (GRAIN WIDTH 2)*, *GL3.1 (GRAIN LENGTH 3.1)*, *GL7 (GRAIN LENGTH 7)*, *GW5 (GRAIN WIDTH 5)*, *GLW7 (GRAIN LENGTH AND WEIGHT ON CHROMOSOME 7)*, *GW8 (GRAIN WIDTH 8)*, and *GIF7* (*GRAIN INCOMPLETE FILLING 7*) have been identified that are involved in the cell division mechanism to control grain shape and width subjected to selection during the domestication process [16,17,18,19,20,21,22,23,24,25].

Two new AA genome *Oryza* species have been currently recognized from northern Australia: an annual and perennial *Oryza meridionalis* and a perennial *Oryza rufipogon*-type taxa [26]. A phylogenetic study based on restriction fragment length polymorphism (RFLP) and short interspersed elements (SINE) data revealed Australian *O. rufipogon* was closely related to Asian *Oryza rufipogon* Griff. and *Oryza nivara* Sharma et Shastry and subsequently formed a cluster together with cultivated *O. sativa*, while *O. meridionalis* is genetically distinct from Asian *O. rufipogon* [27]. That might have happened due to the split of *O. meridionalis* from the other *Oryza* species around 4.8–2.41 million years ago (Ma.) [28,29], and then from Asian *O. rufipogon* approximately between 0.4 and 2 Ma. in the Early Pliocene, long before the domestication of *O. sativa* and the dispersal of *O. rufipogon* to Australia [30]. These two Australian wild rice species are geographically isolated and believed to have no genetic exchange with domesticated rice [26]. The analysis of domestication loci is so far restricted to only *O. rufipogon* and *O. nivara* populations with few domestication genes [4,15,31]. Recent nuclear and chloroplast data showed that the two Australian wild rice, namely, *O. rufipogon*-type taxa, and *O. meridionalis* are novel taxa of the AA genome clade that includes domesticated rice [32]. The former species has a nuclear genotype that is similar to Asian *O. rufipogon*, while *O. meridionalis* Ng. represents a divergent gene pool [33]. Thus, these Australian wild rice populations should be ideal to exploit for the analysis of domestication loci. Recently, hybrids between the *O. rufipogon* type and *O. meridionalis* were observed in nature, indicating the ongoing reticulate evolution of these species [34].

We aimed to determine the sequence variability of domestication loci in wild rice populations determined as single nucleotide polymorphisms (SNPs) relative to *O. sativa* ssp. *japonica* cv. Nipponbare.

## 2. Results

### 2.1. The Output of Mapped Reads

The alignment of reads to the common high-quality *O. sativa* genome (Os-Nipponbare-Reference-IRGSP-1.0) exhibited a considerable sequence variation among the 26 samples (Appendix A). These samples were comprised of 1 *O. rufipogon*-type taxa (WR24), 20 *O. meridionalis,* and 5 hybrids between these two species found in the wild (WR44, WR62, WR52, WR153, and WR161). The average sequence read depth (times reference length) varied from 20.7 in WR153 to 46.9 in WR103, while an average mapped sequence read depth (times reference length) varied from 8.88 in WR24 to 46.7 in WR103. The highest and lowest percentages of mapped reads were observed in WR44 (72.0%) and WR24 (24.0%), respectively. Similarly, the highest percentage of mapped bases was 71.9% in WR44 and the lowest in WR24 (23.8%). The total consensus sequences length ranged between 257,716,289 bp (WR37) and 322,074,565 bp (WR24). The total consensus as a percentage of the reference varied from 67.4% (WR37) to 83.0% (WR44). The output of the mapped reads was mentioned in our earlier publication [34], and may reflect the divergence of these species from domesticated rice. The total number of variants varied from 1,918,262 in WR24 to 7,310,928 in WR81 and 87–90% of these variants were SNPs [34].

### 2.2. SNP Variation in Seed Shattering Loci

The total number of SNPs for seed shattering loci varied among the wild rice populations. The number of homozygous SNPs was high compared to the heterozygous SNPs (Figure 1; Appendix A). For the *qSH1* locus, the heterozygous SNPs ranged from 1 to 58 (Figure 1A; Appendix A). The highest number of heterozygous SNPs (58) was determined in WR44, which was an early generation hybrid. Of them, 4 SNPs were non-synonymous (1491C>A, 1073C>G, 1017G>C, and 301C>G) and resulted in amino acid changes Phe497Leu, Ala358Gly, Glu339Asp, and Pro101Ala, respectively. Furthermore, one non-synonymous amino acid change corresponding to heterozygous SNP (72C>A) resulted in His24Gln in *O. meridionalis* and the later generation of the hybrid (WR153). On the contrary, all the samples from *O. meridionalis* had a high number of homozygous SNPs ranging between 32 and 53 compared to the *O. rufipogon*-type taxon (3 SNPs) and early generations of hybrids WR44 (4 SNPs) and WR62 (10 SNPs). The non-synonymous amino acid changes due to homozygous SNP were common in all *O. meridionalis* samples (Appendix A). The maximum number of homozygous non-synonymous base substitutions was 4 (1491C>A, 1073C>G, 1017G>C, 301C>G) leading to amino acid changes Phe497Leu, Ala358Gly, Glu339Asp, and Pro101Ala, respectively, in the *qSH1* locus in *O. meridionalis*.

As with that for *qSH1*, a similar pattern of SNP variation was also observed for both *SH4* (Figure 1B; Appendix A) and *OsSh1*/*SH1* (Figure 1C). The homozygous SNPs were higher in *O. meridionalis* and the later generation of hybrids, while the lowest was observed in the *O. rufipogon*-type taxa and early generation of hybrids for both loci. In the *SH4* locus, two early generation hybrids (WR44 and WR62) showed the highest number of heterozygous SNP, amounting to 22 and 11, respectively. However, only 3 non-synonymous amino acid changes occurred in WR62 (early generation hybrid) by a heterozygous single base substitution in the 729 (G>C), 500 (C>T), and 472 (A>G) base positions resulting in an amino acid change in Gln243His, Ala167Val, and Thr158Ala, respectively (Appendix A). However, 2 amino acid changes were determined in the heterozygous SNPs of later generation hybrid (WR161): 463G>T for Ala155Ser and 455T>C for Val152Ala. In *O. meridionalis*, only 1 heterozygous SNP underwent a non-synonymous amino acid change from Leu to Pro at base position 226 (677T>C) in WR143 and 2 non-synonymous amino acid changes (Leu226Pro for 677T>C and Ala155Ser for 463G>T) were determined in WR265. Non-synonymous amino acid changes due to homozygous SNPs occurred in *O. meridionalis* ranging between 1 and 4 of which important domestication-related amino acid changes from asparagine to lysine occurred at base position 79 (273T>G) located at 34,232,985 bp of chromosome 4 (Figure 2a; Appendix A). For the *SH1*/ *OsSh1* locus, no non-synonymous amino acid change occurred for the heterozygous SNP, while only 1 domestication-related non-synonymous amino acid change occurred for the homozygous SNP in all samples except for WR24, WR44, and WR111, which was Phe24Leu for a base substitution from T to C at base position 70 located at 25,197,521bp on chromosome 1 (Figure 2b).

Unlike *qSH1*, *SH4*, and *SH1*, the *SHAT1* gene exhibited a very low number of heterozygous SNPs in *O. meridionalis* and hybrid populations ranging between 1 and 6 SNPs. (Figure 1D; Appendix A). As a result, no non-synonymous amino acid changes for the heterozygous SNP occurred in all samples, except for WR184, in which a base substitution from C to G at base position 299 resulted in Ala338Gly. The homozygous SNPs varied from 25 (WR111) to 47 (WR195) in *O. meridionalis* and from 35 (WR52) to 40 (WR44) in hybrids. However, only 2 non-synonymous amino acid changes for homozygous SNPs (832G>A for Ala278Thr and 892G>C for Val298Leu) were determined in all the studied species, except for WR62 which accounted for 1 non-synonymous amino acid change (892G>C for Val298Leu). No homozygous and heterozygous SNPs were determined in *O. rufipogon*-type taxa.

### 2.3. SNP Variation in Awn Development Loci

The *AN3-1* gene which controls awn development in rice exhibited a conspicuous SNP variation in Australian wild rice populations (Figure 3A). No SNP was determined in the *O. rufipogon*-type taxa (WR24) for this locus. The number of heterozygous SNPs ranged between 2 in WR44 and 20 in WR161 in hybrid populations, and from 11 in WR280 to 31 in WR184 in *O. meridionalis* populations (Figure 3A; Appendix A). Non-synonymous amino acid changes corresponding to heterozygous SNPs varied from 0 in WR62 to 5 in WR153 in hybrid populations, and from 2 in WR126 to 8 in WR219 in *O. meridionalis* (Figure 3A; Appendix A). The homozygous SNPs varied from 5 in WR44 to 30 in WR52 in hybrid populations, and the corresponding non-synonymous amino acid changes ranged between 1 (WR44) and 3 (WR62, WR153, and WR161). The homozygous SNPs varied from 17 in WR233 to 36 in WR81 in *O. meridionalis* populations, and the corresponding non-synonymous amino acid changes ranged between 1 and 4 (Figure 3A; Appendix A).

The *LABA1* gene showed a different pattern of SNP variation from that observed in *AN3-1* (Figure 3B). Among all accessions, only the early generation hybrid WR62 contained a high number of heterozygous SNPs (39), whereas all other samples had less than 6 and many had none (Appendix A). No non-synonymous amino acid changes for the heterozygous SNP occurred in all the samples. On the contrary, WR62 was likely to exhibit a smaller number of homozygous SNPs (1), where the other early generation hybrid WR44 (59) showed a comparable number of homozygous SNPs as *O. meridionalis* (Appendix A). Despite having the high number of homozygous SNPs ranging from 15 in WR233 to 61 in WR81 in *O. meridionalis*, there was no non-synonymous amino acid change corresponding to the homozygous SNPs, except for WR219. Only one single base substitution from G to T at 574 resulted in an amino acid change from alanine to serine at base position 192 in *O. meridionalis* (WR219).

Two genes involved in the regulation of awn elongation (*RAE1* and *RAE2*) showed a quite similar trend of SNP variation (Figure 3C,D). In both loci, the early generation of hybrids exhibited the highest number of heterozygous SNPs, accounted for 52 SNPs in WR44 and 39 SNPs in WR62 for the *RAE1* locus, and 11 SNPs in WR44 and 14 SNPs for *RAE2* locus. In the *RAE1* locus, the *O. rufipogon*-type taxa showed 19 heterozygous SNPs, while no heterozygous SNPs were determined in *O. meridionalis*, except for WR37 (1 heterozygous SNP) and WR126 (4 heterozygous SNPs) (Appendix A). Similarly, early generation hybrids exhibited a high number of non-synonymous amino acid changes due to heterozygous SNPs in both loci. In the *RAE1* locus, 8 non-synonymous amino acid changes due to heterozygous SNPs were found in WR44, followed by 6 in WR62, 4 in *O. rufipogon*-type taxa, and 1 in *O. meridionalis* (WR126) (Appendix A). Furthermore, only 2 non-synonymous amino acid changes (326G>C for Arg109Pro and 350A>C for *117Ser) were determined in the early generation hybrid (WR44) due to a single base substitution in the heterozygous *RAE2* locus. The number of homozygous SNPs was higher in the *RAE1* locus compared to the *RAE2* locus (Figure 3C,D; Appendix A). The number of homozygous SNPs for the *RAE1* locus was lower in early generation hybrids (5 in WR44 and 6 in WR62) compared to later generation hybrids (37 in WR52, 52 in WR161, and 57 in WR153). Furthermore, the homozygous SNPs varied from 14 in WR233 to 59 in WR81 in *O. meridionalis*. A similar pattern of homozygous SNPs for the *RAE2* locus was observed in hybrids and *O. meridionalis* (Figure 3D). In the *O. rufipogon*-type taxa, the number of homozygous SNPs was 3 and 1 for the *RAE1* and *RAE2* locus, respectively. However, only one non-synonymous amino acid change Ala63Gly for 188C<G resulted from the homozygous SNP for the *RAE1* locus in the *O. rufipogon*-type taxa. The number of non-synonymous amino acid changes for homozygous SNPs was also higher in number in the *RAE1* locus than that of the *RAE2* locus in hybrids and *O. meridionalis*. The number of non-synonymous amino acid changes for homozygous SNPs ranged between 2 in early generation hybrids and 13 in WR153 (later generation hybrid) for the *RAE1* locus (Appendix A). In addition, only 2 non-synonymous amino acid changes were observed in the later generation of hybrids for the *RAE2* locus. In *O. meridionalis*, the number of non-synonymous amino acid changes corresponding to homozygous SNPs for the *RAE1* locus ranged between 3 in WR233 and 12 in WR133 and 2 for the *RAE2* locus in all samples except for WR184 (1 non-synonymous amino acid change). It is to be noted that 2 non-synonymous amino acid changes for the *RAE2* locus resulted from both heterozygous and homozygous SNPs in the coding region (326G>C and 350A>C), resulting in amino acid changes from Arg109Pro and *117Ser (* affecting the translational stop codon), respectively.

### 2.4. SNP Variation in Grain Size Loci

The grain size-related genes *GS2* (*GRAIN SIZE 2*) and *GS5* (*GRAIN SIZE 5*) showed a distinct pattern of SNP variation across the 26 samples (Figure 4A,B; Appendix A). For the *GS2* locus, a total of 48 heterozygous SNPs were determined in only the early generation hybrid (WR62), of which 6 SNPs in the coding regions (792G>C, 781G>T, 710C>T, 547G>A, 538T>G, and 7A>C) resulted in non-synonymous amino acid changes: Gln264His, Ala261Ser, Ala237Val, Gly183Ser, Ser180Ala, and Met3Leu, respectively. No heterozygous SNP was determined for both wild rice taxon (Figure 4A; Appendix A). The early generation of hybrids (WR44 and WR62) exhibited a comparatively low number of homozygous SNPs compared to *O. meridionalis* and the later generation of hybrids, representing 13 and 1, respectively (Appendix A). Only 6 homozygous SNPs were accounted for in the *O. rufipogon*-type taxa (WR24) that led to synonymous amino acid changes. The number of homozygous SNPs varied from 29 in WR233 to 71 in both WR207 and WR219 in *O. meridionalis* (Figure 4A; Appendix A). The highest number of non-synonymous amino acid changes (6) due to homozygous SNPs was determined in WR207, WR265, and WR280, and the lowest was in WR233 (1). These base substitutions (792G>C, 781G>T, 710C>T, 605C>G, 547G>A, and 7A>C) resulted in amino acid changes from Gln264His, Ala261Ser, Ala237Val, Ser202Cys, Gly183Ser, and Met3Leu, respectively. Furthermore, 1 common single base substitution (538T>G) resulted in amino acid changes from serine to alanine at base position 180 due to the homozygous SNP in WR44 and heterozygous SNP in WR62.

In the case of the *GS5* locus, the heterozygous and homozygous SNP variation was much lower than that of the *GS2* locus (Figure 4B; Appendix A). Only one heterozygous SNP was found in the *O. rufipogon*-type taxa, while the maximum 2 heterozygous SNPs were found in *O. meridionalis*, followed by 1 in both WR111 and WR287. These heterozygous SNPs resulted in synonymous amino acid changes. As with heterozygous SNPs, the number of homozygous SNPs was also very much lower in both wild rice and hybrid populations, except for one *O. meridionalis* sample (WR207) which exhibited 71 homozygous SNPs. This gene functions to control grain size. It is likely that the larger grain size in the wild populations may reflect some loss of function of this gene. Mutations can accumulate in a gene that is not functioning as they have no impact on fitness, and the large number for WR207 may indicate a complete loss of the gene function. The maximum non-synonymous amino acid changes (3) were observed in *O. meridionalis* (WR207): Met467Ile for 1401G>T, Ile250Val for 748A>G, and Ala72Glu for 215C>A. Apart from these, the other *O. meridionalis* samples and later generation hybrids exhibited one amino acid change (Asp66His) due to a base change from G to C at the base position 196.

The *GW8* (*GRAIN WIDTH 8*) gene corresponds to grain size, shape, and quality in rice [25]. As with *GS2*, the highest number of heterozygous SNPs was observed in the early generation of the hybrid (WR44 = 56), compared to the later generation of the hybrid (WR153 = 1). As with the later generation of hybrids, the number of heterozygous SNPs for *O. meridionalis* was also less, amounting to 4 in WR207; followed by 3 in WR287; and 1 in WR37, WR143, and WR233 (Figure 4C; Appendix A). The number of non-synonymous amino acid changes for heterozygous SNPs varied from 1 in both WR143 and WR233 (Ala278Gly for 833C>G) to 2 in WR207 (Gly395Ala for 1184G>C and Ala400Val for 1199C>T) in *O. meridionalis*. The number of homozygous SNPs was less in the early generation of hybrids (2 in WR44 and 9 in WR62) compared to the later generation of hybrids (40 in WR52, 34 in WR153, and 44 in WR161) and *O. meridionalis* (ranging from 19 to 57) (Figure 4C; Appendix A). Only 1 non-synonymous amino acid change (Met364Val for 1090A>G) occurred in the early generation of hybrids (WR62), while 2 non-synonymous amino acid changes Thr345Ser for 1033A>T and Thr399Ala for 1195A>G were found in the later generation of hybrids and *O. meridionalis*. No homozygous and heterozygous SNPs for the *O. rufipogon*-type taxa were determined for the *GW8* locus.

## 3. Discussion

Understanding the natural variation, in terms of genomic polymorphisms (e.g., SNPs), is crucial for the comprehensive analysis of genes and alleles in plant evolution and environmental adaptation [35]. This study reveals a common trend of SNPs’ variation for seed shattering loci, awn development loci, and grain size loci across 26 Australian wild rice populations. The early generation of hybrids (*O. rufipogon*-type taxa × *O. meridionalis*) shows a high level of heterozygous SNP variation, while homozygous SNP variation prevails in the later generations of hybrids and *O. meridionalis*. This is the expected result, as the homozygous SNP’s position in comparison between the species will be heterozygous at these positions, especially in early generation hybrids. It is noticeable that no or few SNPs are observed in the *O. rufipogon*-type taxa. This may be due to the low percentage of mapped reads unable to extract information of domestication loci. In addition, the small sample size (only 1) may have imposed a limitation. The homozygous loci may result from recurrent backcrossing with one of the parental taxa and become fixed over time in each species. *O. meridionalis* populations diverge from the other species around 4.8–2.41 Ma. [28,29]. These homozygous loci may not have been subject to natural selection. Furthermore, desirable functional polymorphisms are favored in domestication [2]. A homozygous locus for a particular trait remains consistent throughout the successive generations [36]. Therefore, homozygous loci for the domestication traits in *O. meridionalis* populations serve as an ideal source for exploiting these genes for crop improvement.

Seed shattering is one of the key habits of wild rice that have been subjected to a domestication process. Wild rice develops an abscissic layer at the junction of sterile lemma and pedicle, which plays a crucial role in seed shattering. Two loci, *qSH1* and *SH4*, have the largest effect in the regulation of the abscission process compared to the other seed shattering-related loci [37]. This study shows a substantial number of SNPs in the wild populations for these two loci. The two Asian subspecies (*japonica* and *indica*) have a lower level of genetic variation in terms of the nucleotide diversity at both loci than that found in their wild progenitors [38]. This study reveals a functional mutation T273G in *O. meridionalis* exhibiting a wild allele of shattering locus *SH4* (orthologs to *SHA1*), which results in an amino acid change from lysine to asparagine at position +79. Lin et al. [39] divulge that an SNP (g237t) in a dominant shattering gene (*SHA1*) brought about an amino acid change from asparagine to lysine at position +79 in the trihelix DNA-binding domain family of plant transcription factor and a proline-rich region (such as the *MYB* DNA-binding factor) contributing to the reduction of seed shattering in cultivars. Such genetic changes switch off the cell separation process, but not the formation of an abscission zone. This gene is mapped to a 5.5 kb genomic region. Similarly, another seed shattering locus *SH1* (*OsSh1*) also exhibits a wild allele for the seed shattering habit in *O. meridionalis* in the study. A non-synonymous base substitution from T to C at base position 70 results in phenylalanine to leucine at base position 24. A study on seed shattering locus *OsSh1* reveals a missense variant (C to T) at base position +70 for cultivated *japonica* rice; however, a wild allele C is present in Asian *O. rufipogon* [40]. They also postulate that wild rice exhibits high nucleotide diversity at *OsSh1 (SH1)*, while low nucleotide diversity is found in *japonica* rice and moderate diversity in *indica* rice.

Awnlessness is a desirable characteristic during the domestication process. The *O. rufipogon*-type taxa shows shorter awns (4.6–5.0) mm and *O. meridionalis* exhibits longer awns (5.9–14.7) mm, while the hybrids exhibit an awn size from 6.9 mm to 11.8 mm [32]. Two genes, mainly *RAE1* and *RAE2*, have been targeted for artificial selection, and the dysfunctional alleles of these two genes are present in cultivated rice [41]. In this study, the high number of homozygous SNPs for *RAE1* and *LABA1* are determined inferring these loci almost get fixed in all *O. meridionalis* samples compared to the *AN3-1* and *RAE2* locus. Similarly, the Asian wild relatives contain the functional alleles of *RAE1* or *LABA1*, demonstrating wild awn traits [41,42]. This study also shows that an *RAE2* locus exhibits similar homozygous and heterozygous polymorphisms in all samples, inferring incomplete fixation over time. The previous study also postulates that this locus is not completely fixed in cultivated rice (93%) and wild progenitors with dysfunctional alleles of *RAE2* exhibiting the awn elongation trait [41].

Similar to seed shattering and awn development traits, grain size loci also show a high level of genetic variation. The *GS2* locus, which encodes *Growth-Regulating Factor 4* (*OsGRF4*), a transcriptional regulator, increases the grain weight and number in cultivated rice [43]. *GS5* encodes a putative serine carboxypeptidase and *GW8* encodes the SBP-family transcription factor *OsSPL16* positively controlling grain size and shape [17]. The grain size of *O. rufipogon*-type taxa is medium (5.81–5.90 mm), and *O. meridionalis* exhibits long-grain (6.20–6.80 mm) [44]. A previous study shows that natural variation at the *GS5* locus is associated with grain size variation in rice [17]. Therefore, a high level of polymorphism of these loci in Australian wild rice may be exploited to improve grain size, shape, and quality for crop improvement.

## 4. Materials and Methods

### 4.1. Sample Collection, DNA Extraction, Quality Check, and Genome Sequencing

The fresh leaves of 26 wild rice populations were collected in the two consecutive years 2015 and 2016 by Moner et al. [32]. Of these, 22 samples were collected in 2015 from 22 different geographical sites in the northern parts of Cape York Peninsula, and only 4 samples were collected in 2016 from the south of Townsville. These samples were comprised of 1 *O. rufipogon*-type taxa (WR24), 20 *O. meridionalis,* and 5 wild hybrids (WR44, WR62, WR52, WR153, and WR161) (Appendix A). These hybrids were derived from a cross between *O. meridionalis* and *O. rufipogon*-type taxa [34]. Samples were kept in the dry ice all the way from the sampling sites to the laboratory and then kept at −80 °C in the freezer.

Fresh leaves from each sample were pulverized with tissue lyser (Qiagen, Germantown, MD, USA) in liquid nitrogen to prevent thawing throughout the pulverization process. The CTAB method [45] was followed to extract high quality and quantity DNA from pulverized leaf tissues. The quality and quantity of the extracted DNA were assessed with a nanodrop spectrophotometer (Thermo Fisher Scientific, Delaware, USA) and electrophoresis on a 0.7% agarose gel stained with SYBR safe. The 260/280 nm ratio ranged from 1.11 to 2.06. Nextera DNA Flex Libraries were prepared that were sequenced on a NovaSeq 6000 SP and S4 Flow cell along with other samples to produce 2 × 150 paired end reads with the data yield of 20× whole-genome coverage on average. A quality check (QC) of the raw data was assessed with CLC Genomic Workbench version 20 software (Qiagen CLC Genomics Workbench 20, Aarhus, Denmark) to identify any sequencing errors before performing further analysis. Sequences were trimmed at both 0.05 and 0.01 quality scores to truncate low quality reads. The 0.01 trimmed reads were considered for further analysis based on the percentage of loss of reads and bases. The total Illumina raw reads and nucleotides for the 26 samples varied from 56,349,078 to 104,325,654, and 8,416,673,405 to 15,628,960,467, respectively. The total 0.01 trimmed reads and nucleotides varied from 54,631,067 to 98,924,768 and 7,925,007,230 to 14,260,593,039, respectively (Appendix A, [34]).

### 4.2. Mapping Query Reads to Reference and Basic Variants’ Detection

A local alignment of the 0.01 trimmed data to a reference Os-Nipponbare-Reference-IRGSP-1.0 [46] was made using CLC Genomic Workbench version 20 software. To filter out the best alignment for reads to the reference, the three different alignment quality thresholds 1.0 length fraction (LF) and 0.95 similarity fraction (SF), 1.0 LF and 0.90 SF, and 1.0 LF and 0.85 SF were applied. In comparison with the results of average mapped reads’ depth (times of reference length), % mapped reads, % mapped bases, total consensus length as % reference length, and total heterozygous variants frequency, the 1.0 LF and 0.90 SF mapping quality threshold with the 25% minimum heterozygous frequency parameter was selected for final mapping. Trimmed reads were mapped to the reference genome with the following parameters: masking mode = no masking; match score = 1; mismatch cost = 2; cost of insertions and deletions = linear gap cost; insertion cost = 3; deletion cost = 3; length fraction = 1.0; similarity fraction = 0.9; global alignment = no; auto-detected paired distances = yes; non-specific match handling = map randomly.

Mapped reads were used to determine the basic variant analysis using the following parameters via CLC Genomic Workbench version 20 software: Ploidy = 2; ignore positions with coverage above = 100,000; restrict calling to target regions = not set; ignore broken pairs = no; ignore non-specific matches = no; minimum coverage = 10; minimum count = 3; minimum frequency (%) = 25; base quality filter = yes; base quality filter = yes; neighborhood radius = 5; minimum central quality = 20; minimum neighborhood quality = 15; read direction filter = no; read position filter = no; remove pyro-error variants = no.

### 4.3. Analysis of Domestication Loci

Genome-wide single nucleotide polymorphisms (SNPs) were determined for seed shattering loci, awn development loci, and grain size loci. Four major genes controlling seed shattering including *qSH1* (*Shattering (QTL)-1*, *SH4/SHA1* (*SHATTERING 4*), *SHAT1* (*SHATTERING ABORTION 1*), and *SH1/OsSh1* (*SHATTERING 1*); four awn development loci including *AN3-1 (Awn3-1), LABA1* (*LONG AND BARBED AWN 1*), *RAE1* (*REGULATOR OF AWN ELONGATION 1*), and *RAE2 (REGULATOR OF AWN ELONGATION 2*); and three loci related to grain size (*GRAIN SIZE 2*; *GS2*)*,* (*GRAIN SIZE 5*; *GS5*), and (*GRAIN WIDTH 8*; *GW8)* were investigated. The structure and functions of domestication loci were extracted from the rice reference genome (Os-Nipponbare-Reference-IRGSP-1.0) downloaded at Rice Annotation Project Database (RAP-DB) [46]. The details of the sequences extracted are given in Appendix A. Single nucleotide polymorphisms (SNPs) across the whole genome were determined with CLC Genomic Workbench version 20 software. These two sets of genomic features (e.g., basic variant track with SNPs and targeted domestication gene regions) were overlapped to screen out domestication gene regions from the basic variant tracks using *.bedtools intersect* command line with BEDtools version 2.30.0 [47]. An amino acid change analysis was conducted in CLC Genomic Workbench version 20 software. Then, heterozygous and homozygous SNPs were filtered based on the frequency parameters: 25–75% frequency as “heterozygous SNP” and 100% frequency as “homozygous SNP”.

## 5. Conclusions

Natural polymorphisms in domestication loci (e.g., seed shattering, awn development, and grain size) are rich in the Australian wild rice populations. The homozygosity of the polymorphisms at these loci confirms the distinctness of the alleles and suggests that these taxa may be ideal for an analysis of domestication.

## Figures and Tables

**Figure 1 plants-12-00489-f001:**
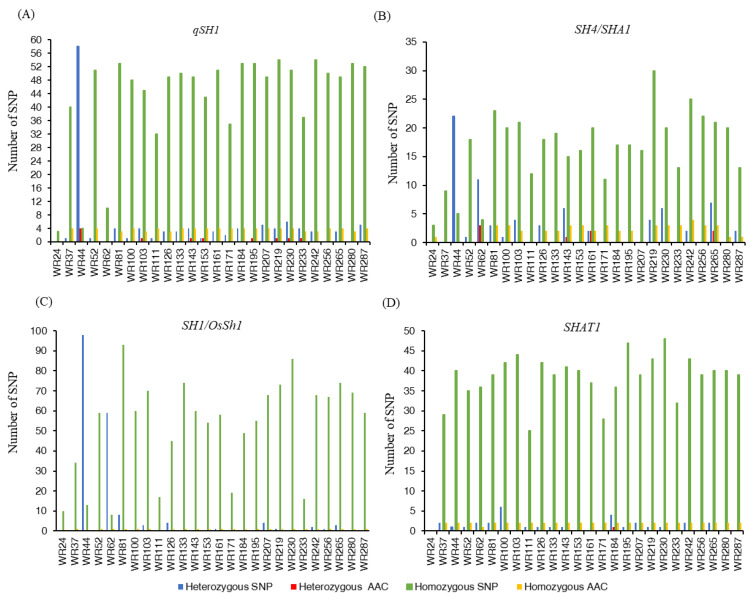
Variation of homozygous and heterozygous single nucleotide polymorphisms (SNPs) and corresponding amino acid changes (AAC) of four seed shattering loci in 26 Australian wild rice samples: (**A**) Total number of homozygous and heterozygous single nucleotide polymorphisms (SNPs) and corresponding amino acid changes (AAC) of *qSH1* (*Shattering (QTL)-1*); (**B**) Total number of homozygous and heterozygous single nucleotide polymorphisms (SNPs) and corresponding amino acid changes (AAC) of *SH4*/*SHA1* (*SHATTERING 4*); (**C**) Total number of homozygous and heterozygous single nucleotide polymorphisms (SNPs) and corresponding amino acid changes (AAC) of *SH1*/*OsSh1* (*SHATTERING 1*); (**D**) Total number of homozygous and heterozygous single nucleotide polymorphisms (SNPs) and corresponding amino acid changes (AAC) of *SHAT1* (*SHATTERING ABORTION 1*).

**Figure 2 plants-12-00489-f002:**
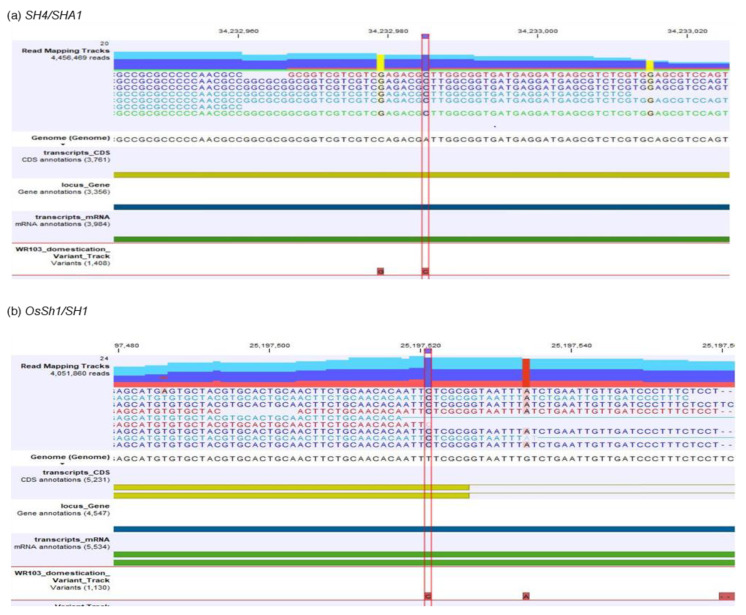
An alignment track comprising read mapping tracks of *O. meridionalis* sample (WR103) along with a reference genome, coding sequence annotation, gene annotation, mRNA annotation, and single nucleotide polymorphism track of WR103: (**a**) an alignment track of *SHATTERING 4* (*SH4*/*SHA1*); (**b**) an alignment track of *SHATTERING 1* (*OsSh1*/*SH1*).

**Figure 3 plants-12-00489-f003:**
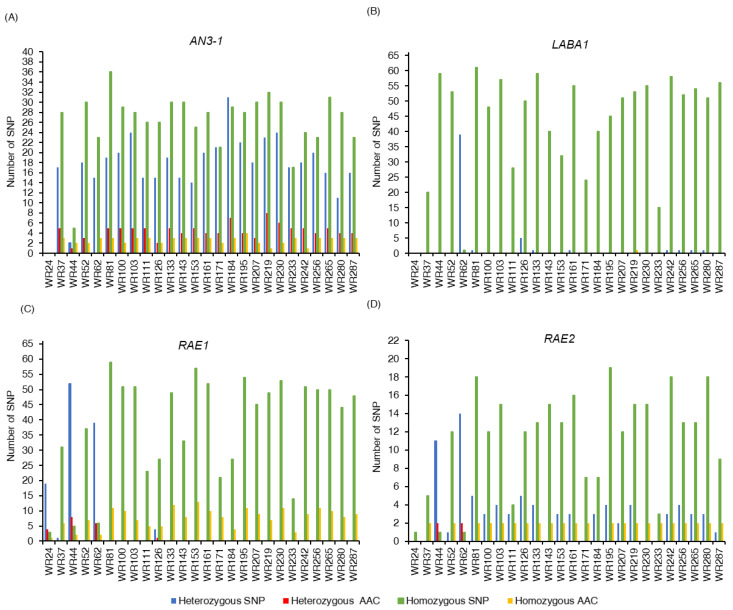
Variation of homozygous and heterozygous single nucleotide polymorphisms (SNPs) and corresponding amino acid changes (AAC) of four loci related to awn development in 26 Australian wild rice samples: (**A**) Total number of homozygous and heterozygous single nucleotide polymorphisms (SNPs) and corresponding amino acid changes (AAC) of *AN3-1* (*Awn3-1*); (**B**) Total number of homozygous and heterozygous single nucleotide polymorphisms (SNPs) and corresponding amino acid changes (AAC) of *LABA1* (*LONG AWN AND BARB 1*); (**C**) Total number of homozygous and heterozygous single nucleotide polymorphisms (SNPs) and corresponding amino acid changes (AAC) of *RAE1* (*REGULATION OF AWN ELONGATION 1*); (**D**) Total number of homozygous and heterozygous single nucleotide polymorphisms (SNPs) and corresponding amino acid changes (AAC) of *RAE2* (*REGULATION OF AWN ELONGATION 2*).

**Figure 4 plants-12-00489-f004:**
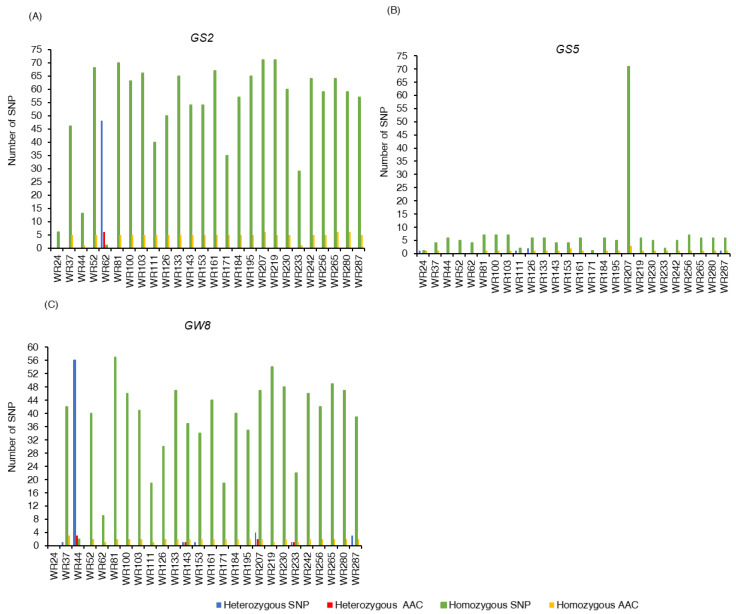
Variation of homozygous and heterozygous single nucleotide polymorphisms (SNPs) and corresponding amino acid changes (AAC) of three grain size loci in 26 Australian wild rice samples: (**A**) Total number of homozygous and heterozygous single nucleotide polymorphisms (SNPs) and corresponding amino acid changes (AAC) of *GS2* (*GRAIN SIZE 2*); (**B**) Total number of homozygous and heterozygous single nucleotide polymorphisms (SNPs) and corresponding amino acid changes (AAC) of *GS5* (*GRAIN SIZE 5*); (**C**) Total number of homozygous and heterozygous single nucleotide polymorphisms (SNPs) and corresponding amino acid changes (AAC) of *GW8* (*GRAIN WIDTH 8*).

## Data Availability

The datasets presented in this study can be found in online repositories. The names of the repository/repositories and accession number(s) and date can be found below: https://www.ncbi.nlm.nih.gov/genbank/, BioProject number PRJNA758754 (accessed on 22 February 2022).

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
