# Peer review of "Analysis of Domestication Loci in Wild Rice Populations"

_plants, 2023, doi:10.3390/plants12030489_

Round 1
Reviewer 1 Report
The authors intended to conduct a study for the investigation of domestication-related loci based on SNP variations in a wild rice species Oryza meridionalis. The topic sounds interesting while the scientific significance from the results is unclear a bit. The manuscript looks descriptive and might need an impressive detailed presentation about the representative locus/allele preferably.
(1) About the calculation design
Two-types of polymorphic alleles, “homozygous SNPs” and “heterozygous SNPs”, were designated in the variant analysis. What are these exactly? SNPs between the O. sativa reference alleles and the sample alleles? Or SNPs among the O. meridionalis population? What are the five hybrid lines? Are these related to O. meridionalis?
The numbers of SNPs on each domestication gene were presented on the manuscript. Among the SNPs, non-synonymous substitutions were highlighted, but their biochemical functionality remains described more. How are the amino acid properties? And how the non-synonymous substitutions are related to the variances in domesticated rice?
What about the difference between the genic SNPs and the intergenic (in particular promoter SNPs)? Promoter SNPs may affect the rice trait.
There is a good example to present the SNP information of domestication genes.
Yamamoto et al. Rice (2018) 11:33
https://doi.org/10.1186/s12284-018-0224-3
(2) Relationship among the analyzed domestication loci
The present manuscript shows the SNP variations of the analyzed domestication genes separately. How was the relationship of SNPs among these genes?
(3) The body structure
Materials and Methods were placed after the Discussion section. Is the current structure correct?
Author Response
Reviewer 1:
The authors intended to conduct a study for the investigation of domestication-related loci based on SNP variations in a wild rice species Oryza meridionalis. The topic sounds interesting while the scientific significance from the results is unclear a bit. The manuscript looks descriptive and might need an impressive, detailed presentation about the representative locus/allele preferably.
(1) About the calculation design
Two-types of polymorphic alleles, “homozygous SNPs” and “heterozygous SNPs”, were designated in the variant analysis. What are these exactly? SNPs between the O. sativa reference alleles and the sample alleles? Or SNPs among the O. meridionalis population? What are the five hybrid lines? Are these related to O. meridionalis?
Response: The homozygous and heterozygous SNP of O. rufipogon type taxa, O. meridionalis and hybrids were determined in relation to O. sativa. These SNPs were between the O. sativa reference alleles and the sample alleles. The five hybrid lines were naturally occurring hybrids between O. rufipogon type taxa and O. meridionalis. The following text is added in line 88-89 in the manuscript
Recently, hybrids between O. rufipogon type and O. meridionalis were observed in nature suggesting ongoing reticulate evolution [34].
The numbers of SNPs on each domestication gene were presented on the manuscript. Among the SNPs, non-synonymous substitutions were highlighted, but their biochemical functionality remains described more. How are the amino acid properties? And how the non-synonymous substitutions are related to the variances in domesticated rice?
Response: the objective of this study was to determine SNP variation. Non-synonymous substitution results in amino acid changes described in the manuscripts. These SNPs in wild rice were determined compared to O. sativa. Protein sequence cannot be used to reliably predict function.
What about the difference between the genic SNPs and the intergenic (in particular promoter SNPs)? Promoter SNPs may affect the rice trait.
Response: The total number of SNPs in domestication genes were determined and SNP at the whole genome level were assessed in the study. Many SNP outside the coding region may influence traits.
There is a good example to present the SNP information of domestication genes.
Yamamoto et al. Rice (2018) 11:33
https://doi.org/10.1186/s12284-018-0224-3
(2) Relationship among the analyzed domestication loci
The present manuscript shows the SNP variations of the analyzed domestication genes separately. How was the relationship of SNPs among these genes?
Response: Determination of teh sequence of haplotypes was not possible in these SNP data
(3) The body structure
Materials and Methods were placed after the Discussion section. Is the current structure correct?
Response: The template of this journal was followed. So, the structure of the manuscript is in accordance with the instructions of this journal.
Reviewer 2 Report
This manuscript summarizes the SNPs at several domestication loci in 26 Australian wild rice populations. The authors mapped the Illumina sequencing reads to the O. sativa genome and detected SNP variation at seed shattering loci, Awn development loci, and grain size loci. These SNPs at the domestication loci could be useful as a genetic resource for rice improvement.
The introduction and materials and methods parts are solid. The authors of this study did a great job summarizing the SNPs at domestication loci across 26 wild rice populations. However, this manuscript lacks a full picture of the domestication loci of those wild rice populations. By only showing the SNPs and amino acid changes, it is hard to know what their potential impact is on the protein structure and function of this protein. Potential analysis, such as the prediction of protein structures, could be done to further explore the effect of the SNPs detected in this manuscript. Other kinds of mutations, such as insertion or deletion, should also be reported as well. Ka/Ks ratio could be calculated to show the selection pressure during evolution.
I have the following comments and suggestions:
1. Line 90. All kinds of mutations should be reported in the manuscript, not just SNPs.
2. Line 151. The width of the bars is not consistent within the pictures. The rest histograms have the same issue to be fixed.
3. Line 156. The resolution of this figure is very low and it’s hard to read. The red box should not include the same sites on CDS, gene, and mRNA bars as they vary in length. Are these two SNPs validated in the previous research with their significance in domestication?
4. Line 279. What’s the interpretation of heterozygous vs homozygous SNP and AAC? Most of the loci have much more homozygous SNPs than the heterozygous SNPs, with exceptions like AN3-1 and RAE2.
5. Line 348. What’s the hybrid? Cross between O. rufipogon and O. meridionalis?
Author Response
Reviewer 2:
Comments and Suggestions for Authors
This manuscript summarizes the SNPs at several domestication loci in 26 Australian wild rice populations. The authors mapped the Illumina sequencing reads to the O. sativa genome and detected SNP variation at seed shattering loci, Awn development loci, and grain size loci. These SNPs at the domestication loci could be useful as a genetic resource for rice improvement.
The introduction and materials and methods parts are solid. The authors of this study did a great job summarizing the SNPs at domestication loci across 26 wild rice populations. However, this manuscript lacks a full picture of the domestication loci of those wild rice populations. By only showing the SNPs and amino acid changes, it is hard to know what their potential impact is on the protein structure and function of this protein. Potential analysis, such as the prediction of protein structures, could be done to further explore the effect of the SNPs detected in this manuscript. Other kinds of mutations, such as insertion or deletion, should also be reported as well. Ka/Ks ratio could be calculated to show the selection pressure during evolution.
I have the following comments and suggestions:
- Line 90. All kinds of mutations should be reported in the manuscript, not just SNPs.
Response: the following text is added in line 106-107
The total number of variants varied from 1,918,262 in WR24 to 7,310,928 in WR81 and 87-90% of the variants were SNPs [34].
- Line 151. The width of the bars is not consistent within the pictures. The rest histograms have the same issue to be fixed.
Response: Checked
- Line 156. The resolution of this figure is very low and it’s hard to read. The red box should not include the same sites on CDS, gene, and mRNA bars as they vary in length. Are these two SNPs validated in the previous research with their significance in domestication?
Response: The resolution of this figure is low as it was taken from the CLC-Genomic Workbench. Replaced with a much clearer figure The red box was used to highlight the SNPs. These two SNPs were validated in the previous research by Li et al. 2006 and Li et al. 2020 with their significance in domestication.
- Line 279. What’s the interpretation of heterozygous vs homozygous SNP and AAC? Most of the loci have much more homozygous SNPs than the heterozygous SNPs, with exceptions like AN3-1 and RAE2.
Response: The high number of homozygous loci infer the fixation of alleles in these species over the period of time while heterozygous SNPs in hybrids indicates their recent hybridization. AN3-1 and RAE2 locus exhibit similar both homozygous and heterozygous polymorphisms in all samples inferring incomplete fixation over time.
- Line 348. What’s the hybrid? Cross between O. rufipogonand O. meridionalis?
The following text is added in line 88-89 in the manuscript
Recently, hybrids between O. rufipogon type tax and O. meridionalis were observed in nature suggesting ongoing reticulate evolution [34].
The following text is added in line 401-402
These hybrids were derived from a natural hybrid between O. meridionalis and O. rufipogon type taxa [34].
Reviewer 3 Report
This research article has been investigated on domestication loci based on genomic sequences of genes related to shattering, awnlessness, and grain size in wild rice populations. However, there was not core discovery on domestication. The comments for authors are as follows:
Line 152: In Figure 1 legend, ‘four awn development loci ’ --- This is wrong expression. It needs to be corrected.
Lines 151~155: Figure 1 does not provide important information. Authors may put in a Table or in a Supplementary Figure.
Lines 155~156: Figure 2 is not clear enough to see the distinct information that was explained in the text.
Lines 129~140: In O. meridionalis, only 1 heterozygous SNP underwent non-synonymous amino acid change from Leu to Pro at 226 base position (677T>C) in WR143 and 2 non-synonymous amino acid changes (Leu226Pro for 677T>C and Ala155Ser for 463G>T) were determined in WR265. Non-synonymous amino acid changes due to homozygous SNPs occurred in O. meridionalis ranging between 1 and 4 of which one important domestication related amino acid changes from asparagine to lysine occurred at 79 base position (273T>G) located at 34,232,985 bp of chromosome 4 (Figure 2a; Table S3). For SH1/ OsSh1 locus, no non-synonymous amino acid change occurred for heterozygous SNP while only 1 domestication related non-synonymous amino acid change occurred for homozygous SNP in all samples except for (WR24, WR44, and WR111), which was Phe24Leu for a base substitution from T to C at 70 base position located at 25,197,521 base position on chromosome 1 (Figure 2b). --- Authors did not provide the clear evidences on domestication of SH1 locus.
Lines 217~227: Figure 3 does not provide important information. Authors may put in a Table or in a Supplementary Figure.
In the section of “2.3 SNP variation in Awn development loci”, authors did not provide the meaningful SNPs. They just described about the SNP variations.
Lines 273~274: Figure 4 does not provide important information. Authors may put in a Table or in a Supplementary Figure.
Overall, this article does not provide the important results on domestication of shattering, awnlessness, and grain size.
Therefore, this article is not suitable for publication in the Plants journal.
Author Response
Reviewer 3:
Comments and Suggestions for Authors
This research article investigated domestication loci based on genomic sequences of genes related to shattering, awnlessness, and grain size in wild rice populations. However, there was not core discovery on domestication. The comments for authors are as follows:
Line 152: In Figure 1 legend, ‘four awn development loci ’ --- This is wrong expression. It needs to be corrected.
Response: corrected
Lines 151~155: Figure 1 does not provide important information. Authors may put in a Table or in a Supplementary Figure.
Response: A table is now provided in the supplementary as Table S2
Lines 155~156: Figure 2 is not clear enough to see the distinct information that was explained in the text.
Response: the resolution of this figure is low as it was taken from the CLC-Genomic Workbench. Now replaced with a much clearer figure.
Lines 129~140: In O. meridionalis, only 1 heterozygous SNP underwent non-synonymous amino acid change from Leu to Pro at 226 base position (677T>C) in WR143 and 2 non-synonymous amino acid changes (Leu226Pro for 677T>C and Ala155Ser for 463G>T) were determined in WR265. Non-synonymous amino acid changes due to homozygous SNPs occurred in O. meridionalis ranging between 1 and 4 of which one important domestication related amino acid changes from asparagine to lysine occurred at 79 base position (273T>G) located at 34,232,985 bp of chromosome 4 (Figure 2a; Table S3). For SH1/ OsSh1 locus, no non-synonymous amino acid change occurred for heterozygous SNP while only 1 domestication related non-synonymous amino acid change occurred for homozygous SNP in all samples except for (WR24, WR44, and WR111), which was Phe24Leu for a base substitution from T to C at 70 base position located at 25,197,521 base position on chromosome 1 (Figure 2b). --- Authors did not provide the clear evidences on domestication of SH1 locus.
Response: the following information is already mentioned in the manuscript regarding the domestication of SH1/OsSh1 locus.
“only 1 domestication related non-synonymous amino acid change occurred for homozygous SNP in all samples except for (WR24, WR44, and WR111), which was Phe24Leu for a base substitution from T to C at 70 base position located at 25,197,521 base position on chromosome 1”. The impact of this SNP in domestication was validated by Li et al. 2020.
Lines 217~227: Figure 3 does not provide important information. Authors may put in a Table or in a Supplementary Figure.
Response: A table is now provided in the supplementary as Table S5
In the section of “2.3 SNP variation in Awn development loci”, authors did not provide the meaningful SNPs. They just described the SNP variations.
Response: Awn development loci were not the result of SNPs related to domestication. Therefore, we just described the variation of SNPs for the four loci related to awn development. Differences in regulatory genes may explain awn morphology.
Lines 273~274: Figure 4 does not provide important information. Authors may put in a Table or in a Supplementary Figure.
Response: A table is now provided in the supplementary as Table S8
Overall, this article does not provide the important results on domestication of shattering, awnlessness, and grain size.
Therefore, this article is not suitable for publication in the Plants journal.
Response: The article determines for the first time the sequence variation of the key domestication genes in wild rice that has not been impacted by domestication or gene flow from domesticated populations. Unlike Asian populations, the Australain populations have been isolated from the impact of interbreeding with the large domesticated population is Asia for thousands of years. It shows that the changes are complex and not yet fully understood.
Round 2
Reviewer 1 Report
Based on the authors’ response to my comments, the revised manuscript was evaluated. This study predicted SNP variations of domestication-related loci in a population for the wild rice species Oryza meridionalis. The revised manuscript is bit better to read than the original one. However, the reviewer still thinks that this study is quite descriptive and might need an impressive detailed presentation about the representative locus/allele preferably to highlight scientific significance. Several comments were left as follows. Hope all of issues will be addressed well in the next revision.
In Line 90-92, the authors described the aim of this study: “determine sequence variability of domestication loci in wild rice population…”. Here are two issues to be addressed; (1) candidate SNP sites looked not validated experimentally, and (2) it is unclear what is the scientific significance given from the determination of SNP variations on domestication genes in the present research design. The reviewer supposes that the investigation of SNP variations would provide a data resource, but at least one good example (known functional SNP site/haplotype or potentially functional SNP variations, etc.) with valid biological significance should be clarified among them. In case the authors insist on only showing SNP variation information, valid control data set should be added to show differences between the control and O. meridionalis. There might be a way: for example, adding additional O. rufipogon lines, or calculating the relative SNP variability in each domestication gene based on whole genome SNPs.
In Line 97-108, the authors described the output of mapped reads. Here is a question raised. What were done in their previous study, and what were done for this study exactly? This is unclearly written in the revised manuscript. Things done in the previous study should be briefly described with the reference and should not be redundantly reported between two studies.
According to the authors’ response, the candidate SNP sites on genic and intergenic regions (including promoter region) looked analyzed. However, it is unclear in the text that which regions were analyzed exactly. This is due to lacking of detailed documentation in Methods. The reviewer guesses that the analyzed regions would comprises the full coding regions, all the intronic regions, 5’ UTR and 3’ UTR regions, and probable promoter regions (? kb upstream from the translation start site or predicted transcription start site).
In Line 469-471, the authors designated the heterozygous and homozygous SNPs based on the allelic frequency parameter. The reviewer understands that 25-75% was of heterozygous SNPs, and 100% was of homozygous SNPs. Then what about 75-100% SNPs?
Author Response
In Line 90-92, the authors described the aim of this study: “determine sequence variability of domestication loci in wild rice population…”. Here are two issues to be addressed; (1) candidate SNP sites looked not validated experimentally, and (2) it is unclear what is the scientific significance given from the determination of SNP variations on domestication genes in the present research design. The reviewer supposes that the investigation of SNP variations would provide a data resource, but at least one good example (known functional SNP site/haplotype or potentially functional SNP variations, etc.) with valid biological significance should be clarified among them. In case the authors insist on only showing SNP variation information, valid control data set should be added to show differences between the control and O. meridionalis. There might be a way: for example, adding additional O. rufipogon lines, or calculating the relative SNP variability in each domestication gene based on whole genome SNPs.
Response: The manuscript details the functional SNP discovered in the wild populations for each of the loci studied. For example important variation in the shattering loci was found in the wild populations. These are mostly O meridionalis samples (all shattering) and they were analysed against a reference (control) O. sativa genome.
In Line 97-108, the authors described the output of mapped reads. Here is a question raised. What were done in their previous study, and what were done for this study exactly? This is unclearly written in the revised manuscript. Things done in the previous study should be briefly described with the reference and should not be redundantly reported between two studies.
Response: This study is an analysis of data that has been published in an earlier study of the samples that focused on understanding the relationships between the wild accessions. In this present study the domestication loci have been studied and it is necessary to cite the earlier wok as the source of the primary data.
According to the authors’ response, the candidate SNP sites on genic and intergenic regions (including promoter region) looked analyzed. However, it is unclear in the text that which regions were analyzed exactly. This is due to lacking of detailed documentation in Methods. The reviewer guesses that the analyzed regions would comprises the full coding regions, all the intronic regions, 5’ UTR and 3’ UTR regions, and probable promoter regions (? kb upstream from the translation start site or predicted transcription start site).
Response: The reviewer is correct, only the coding regions were analysed.
In Line 469-471, the authors designated the heterozygous and homozygous SNPs based on the allelic frequency parameter. The reviewer understands that 25-75% was of heterozygous SNPs, and 100% was of homozygous SNPs. Then what about 75-100% SNPs?
The calling of heterozygous and homozygous SNP used these conventional conservative limits to ensure no false calls. The very small numbers of 75-100% were not classified.
Response:
Reviewer 2 Report
In the revised version, the authors addressed some detailed issues. However, this manuscript still lacks depth and misses the big picture. As previously suggested, by only listing the SNPs and amino acid changes, it is hard to know what their potential impact is on the protein structure and function of this protein. Potential analysis, such as the prediction of protein structures, could be done to further explore the effect of the SNPs detected in this manuscript. Other kinds of mutations, such as insertion or deletion, should also be reported as well. Ka/Ks ratio could be calculated to show the selection pressure during evolution.
Author Response
In the revised version, the authors addressed some detailed issues. However, this manuscript still lacks depth and misses the big picture. As previously suggested, by only listing the SNPs and amino acid changes, it is hard to know what their potential impact is on the protein structure and function of this protein. Potential analysis, such as the prediction of protein structures, could be done to further explore the effect of the SNPs detected in this manuscript. Other kinds of mutations, such as insertion or deletion, should also be reported as well. Ka/Ks ratio could be calculated to show the selection pressure during evolution.
Response: Prediction of protein function from structure is difficult and unreliable. The diversity of protein sequences discovered here could best be tested in long term research in transgenic plants utilizing the outcomes of this current study. Insertions and deletions are much less common (10 times) in these genomes relative to O. sativa as we have reported earlier. We do not have evidence for indels that could be important in the domestication genes.